# A Simple and Cost-Efficient Platform for a Novel Porcine Circovirus Type 2d (PCV2d) Vaccine Manufacturing

**DOI:** 10.3390/vaccines11010169

**Published:** 2023-01-12

**Authors:** Sarawuth Noppiboon, Neeracha Lapanusorn, Pisit Ekkpongpaisit, Sarah Slack, Stefanie Frank, Lalintip Hocharoen

**Affiliations:** 1Bioprocess Research and Innovation Centre, King Mongkut’s University of Technology Thonburi, Bangkok 10140, Thailand; 2Department of Biochemical Engineering, University College London, Gower Street, London WC1E 6BT, UK

**Keywords:** large-scale, fed-batch fermentation, one-step purification, porcine circovirus, veterinary vaccine

## Abstract

Porcine circovirus type 2d (PCV2d) is becoming the predominant PCV genotype and considerably affects the global pig industry. Nevertheless, currently, no commercial PCV2d vaccine is available. Preventing and controlling the disease caused by PCV2d is therefore based on other genotype-based vaccines. However, their production platforms are laborious, limited in expression level, and relatively expensive for veterinary applications. To address these challenges, we have developed a simple and cost-efficient platform for a novel PCV2d vaccine production process, using fed-batch *E. coli* fermentation followed by cell disruption and filtration, and a single purification step via cation exchange chromatography. The process was developed at bench scale and then pilot scale, where the PCV2d subunit protein yield was approximately 0.93 g/L fermentation volume in a short production time. Moreover, we have successfully implemented this production process at two different sites, in Southeast Asia and Europe. This demonstrates transferability and the high potential for successful industrial production.

## 1. Introduction

Porcine circovirus type 2 (PCV2) is one of the most pervasive viral pathogens repeatedly affecting the swine meat production industry worldwide, as infection causes loss in pig production, mostly due to growth retardation, reduction in average daily weight gain, and reproductive issues. Importantly, this leads to immune suppression, increasing the risk of other pathogenic infections. PCV2 is also associated with the occurrence of a postweaning multisystemic wasting syndrome (PMWS), porcine dermatitis and nephropathy syndrome (PDNS), and porcine respiratory disease complex (PRDC) [1,2]. Over the last few years, predominant genotype shifts have been observed; these include a shift from PCV2a to PCV2b in the mid-2000s, and since 2010, a shift to PCV2d [3,4,5,6,7,8]. Currently, this strain is widely circulating in the US [9], China [10], Russia [11], Italy [5], Korea [6], and Thailand [7]. The evolution of PCV2 has raised a debate on cross-protection efficacy. Commercially available PCV2 vaccines are mostly PCV2a-based vaccines which are produced based on inactivated whole PCV2 virus (Circovac^®^) [12], inactivated baculovirus vector (Circumvent^®^ PCV G2) [13], and baculovirus-expressed capsid (cap) protein (PORCILIS^®^) [14,15]. These production technologies are laborious, limited in expression level, and relatively expensive for veterinary applications [16,17,18]. To address these challenges, a simple, reliable, and cost-efficient production process is needed. One attractive choice is using *Escherichia coli* (*E. coli*) microbial fermentation to express a PCV2d subunit vaccine. *E. coli* production can be performed to high cell density cultivation, and the process can be readily conducted on a large scale with a short production process time. Moreover, current processes render yields as high as g/L scales of product [19,20,21].

Recently, a novel PCV2d vaccine and a laboratory-scale production process were developed in order to produce small quantities of test material for animal experiments and vaccine characterization [22]. However, for pig trials and manufacturing purposes, the development of a larger-scale production process is needed. Herein, for the first time, we demonstrate a simple and practical approach for large-scale biomanufacturing of this novel PCV2d subunit vaccine. The designed process covers the fed-batch fermentation at the bench- and pilot-scale, and the product recovery via homogenization, followed by a single-step purification using ion exchange chromatography. This production process has been evaluated by technology transfer to another production site in the UK and was shown to achieve high yields of vaccine products. The successful development of this manufacturing platform enables local production of veterinary vaccines, thereby widening availability and accessibility.

## 2. Materials and Methods

### 2.1. Fermentation

#### 2.1.1. *E. coli* Cell Banks and Preculture in Shake Flasks

Cell stocks of *E. coli* W3110 containing pET23/ptac *Δ2–40* PCV2d-His6 were received from the Robinson group at the University of Kent (UK) [22]. Research cell banks (RCB) were prepared by growing cells to an OD600 of 0.5–0.7 in Terrific Broth (TB) (BD Difco^TM^, Sparks, NV, USA) containing 5 g/L glycerol and 100 µg/mL ampicillin at 30 °C and 200 rpm in a shaking incubator (Innova43R, Eppendorf New Brunswick, Enfield, CT, USA). Cells were mixed with 40% *w*/*v* glycerol in a 1:1 ratio and stored at −80 °C for use in this whole study.

The preculture was started with 1% inoculation of RCB to 50 mL TB in a shake flask and incubated at 30 °C and 200 rpm overnight. For cell expansion to a pilot-scale fermenter, the first preculture was inoculated into a second preculture with 10% cell density in 200 mL SM6Gc media supplemented with antibiotics and cultured at 30 °C and 200 rpm until OD600 reached 5–7.

SM6Gc comprises SM6 and its trace elements, which were prepared separately. SM6 media consisted of 5.2 g/L (NH_4_)_2_SO_4_, 4.4 g/L NaH_2_PO_4_·H_2_O, 4.03 g/L KCl, 1.04 g/L MgSO_4_·7H_2_O, 4.55 g/L citric acid monohydrate, 0.25 g/L CaCl_2_·2H_2_O, and 95 g/L glycerol. The pH was adjusted to 7.0 using 25% NH_4_OH. The trace elements were prepared at 100× stock solutions containing 113.48 g/L citric acid monohydrate, 5.22 g/L CaCl_2_·2H_2_O, 2.06 g/L ZnSO_4_·7H_2_O, 2.028 g/L MnSO_4_·H_2_O, 0.81 g/L CuSO_4_·5H_2_O, 0.42 g/L CoSO_4_·7H_2_O, 10.06 g/L FeCl_3_·6H_2_O, 0.03 g/L H_3_BO_3_, and 0.02 g/L Na_2_MoO_4_·2H_2_O. The trace element solutions were filtered through a 0.22 µm sterile syringe filter and stored in dark bottles at 4 °C. Filters and chemicals were all purchased from Merck KGaA(Darmstadt, Germany) unless otherwise stated.

#### 2.1.2. Fed-Batch Cultivations in a Stirred-Tank Fermenter

A 3.7 L Ralf Advanced (Bioengineering AG, Wald, Switzerland) stirred-tank glass fermenter, equipped with a 2-flat-blade Rushton agitator, a 6-blade agitator for radial mixing, 4 stainless steel baffles, and a ring sparger, was used in this work. A quantity of 1.5 L of SM6 media was transferred to the fermenter followed by autoclaving at 121 °C for 20 min (VE150, Systec GmbH, Linden, Germany). An amount of 15 mL 100× trace elements solution was filter-sterilized via a 0.22 µm syringe filter into the vessel before inoculation. The pH set point of 7.0 was controlled by a cascade using internal peristaltic pumps for the addition of 10% H_3_PO_4_ or 25% NH_4_OH. The pH dead band was set to 0.1. Dissolved oxygen (DO) concentration (%DO) was calibrated at 0 by unplugging the probe’s cable. Calibration of 100% DO was achieved by keeping the mixing speed at 1050 rpm and the airflow rate at 60 Nl/h. The %DO setpoint was 30 and was controlled by a cascade of agitation speeds (450–1050 rpm), airflow, and O_2_ flow. A total gas flow of 1 vvm (volume of gas sparged per fermenter working volume per minute) was used. The starting temperature was set at 30 °C and was reduced to 25 °C once the cells reached an OD600 of 60–70. The temperature was controlled through perfused stainless-steel baffles connected to the heating circuit with a circulation pump, electrical heater, and cooling water valve. In addition, 5% Antifoam 204 (Sigma-Aldrich, Inc., St. Louis, MO, USA) was used to prevent overflow of the media during cultivation through control of the level probe and peristaltic pump.

Fermentation was performed with a starting OD600 of around 0.8. A series of supplements were added through a 0.22 µm sterile syringe filter as follows: 1.972 g/L MgSO_4_·7H_2_O when OD600 reached 38–42, 1.164 g/L NaH_2_PO_4_·H_2_O at OD600 of 54–58 and 1.63 g/L NaH_2_PO_4_·H_2_O when starting the fed-batch process at OD600 around 66–70. The feeding media, 800 g/L glycerol, was fed constantly at 0.15 mL/min (per 1.5 L working volume) using the internal pump. When OD600 exceeded 75, cells were induced with 0.039 g/L Isopropyl β-d-1-thiogalactopyranoside (IPTG). Cells were grown until an OD600 of 200 was reached, then the temperature was lowered to 15 °C prior to proceeding to cell harvest by centrifugation at 7500× *g* for 30 min (LYNX 6000, Thermo Fisher Scientific, Osterode am Harz, Germany). Wet cell paste was collected and stored at −20 °C for downstream processing. Fermentation samples were collected throughout the cultivation for OD600 measurements. A 1 mL cell suspension was taken, before and after induction, and spun down, and the pellet was kept for SDS-PAGE and Western blot analysis.

#### 2.1.3. Fed-Batch Cultivations in a Stirred-Tank Pilot-Scale Fermenter

Fed-batch processes at pilot scale were performed in a 42 L stainless-steel stirred-tank bioreactor (BIOSTAT^®^ D-DCU, Sartorius Stedim Biotech GmbH, Guxhagen, Germany) with 4 stainless-steel baffles and 3 six-bladed Rushton turbines. A quantity of 20 L of SM6 media was prepared and transferred to the fermenter through an external peristaltic pump. Then, automatic sterilization in place (SIP) was conducted using a recipe pre-programmed in BioPAT^®^ MFCS|win. After cooling down, 200 mL 100X trace element solution was added to the sterile media using a 0.2 µm sterile disc filter (Sartolab^®^ P20, Sartorius AG, Goettingen, Germany). 

The operating conditions were set as described above with a small modification in %DO cascade where agitation ranged from 250 to 800 rpm while the total gas flow was maintained at 1 vvm. Prior to inoculation, the media was held overnight to verify the sterility. Then, inoculation was carried out using the second preculture to obtain an initial OD600 of around 0.8. Supplements, feeding media, and IPTG were added as described above. The 800 g/L glycerol was fed constantly at 2 mL/min (per 20 L working volume). Fermentation was terminated after cells were grown for more than 48 h and OD600 was over 200. Wet cell paste was collected after centrifugation and stored for future use. Samples were collected and analyzed, as described in the sample preparation and analytical methods section.

### 2.2. Cell Disruption and Filtration

Next, 15% of wet cell paste of *E. coli* W3110 producing PCV2d capsid protein was resuspended in resuspension buffer (50 mM Tris-HCl, 2.5 mM EDTA, pH 7.0) at various NaCl concentrations depending on which CIEX equilibration buffer was used. A disperser was used for mixing at 3000 rpm for 15 min (T50 digital ULTRA-TURRAX^®^, IKA, Staufen, Germany). The resuspended cell sample was lysed by cell disruption using a high-pressure homogenizer (TS Series 4kW, Constant Systems LTD, Northants, UK) at 700, 1400, and 2100 bars for 10 passages until low viscosity of lysed cells was observed. Cell debris was separated by centrifugation at 20,000x *g* for 30 min at 10 °C. The supernatant was collected and filtered through a depth filter (Supracap 50 PDH4, Pall, Bad Kreuznach, Germany) and a 0.45 µm PES membrane disc filter (Supor^®^, Pall, MI, USA). A turbidity meter (L100Q, Hach, Loveland, CO, USA) was used to measure sample clarification before applying the sample onto the chromatographic column.

### 2.3. Cation Exchange Chromatography

Cation exchange chromatography was conducted in a one-step purification (ÄKTA Pure 150, Cytiva, Uppsala, Sweden). A small-scale experiment used 2× HiScreen SP Sepharose Fast Flow columns (Cytiva) connected in series. The linear velocity was kept at 300 cm/h. The binding conditions were optimized based on our preliminary studies (Appendix A) using 50 mM Tris buffer at various concentrations of NaCl (0, 200, and 300 mM). The columns were equilibrated accordingly. Five column volumes (CV) of clarified supernatant from the product recovery step were applied onto the column. Unbound proteins and impurities were first washed with equilibration buffer and subsequently washed with a step gradient at 40% of elution buffer (50 mM Tris-HCl, 1 M NaCl, pH 7.0). PCV2d was eluted at 100% elution buffer. Purification fractions were collected for SDS-PAGE and Western blot.

A HiScale 16/40 column (Cytiva, Uppsala, Sweden) was packed with SP Sepharose Fast Flow resin (Cytiva, Uppsala, Sweden) to 20 cm bed height, requiring 40 mL of resin. Column efficiency was tested according to Cytiva’s protocol. The flow rate was kept constant at a linear velocity of 300 cm/h for the entire run. The column was equilibrated with 50 mM Tris-HCl, 300 mM NaCl, pH 7.0 in 3 column volumes (CV). The clarified supernatant from the product recovery step was applied onto the column for 5 CV using the sample pump. Unbound proteins and impurities were first washed with 4 CV of equilibration buffer and then with a step gradient at 40% of elution buffer (50 mM Tris-HCl, 1 M NaCl, pH 7.0) for another 4 CV. PCV2d was eluted at 100% of elution buffer for 4 CV. Flow through, washing steps, and elution were all collected in 40-, 40-, and 10-mL fractions, respectively. The purified fractions were collected for SDS-PAGE and Bradford analysis.

### 2.4. Analytical Methods

Cell pellets from fermentation runs were resuspended in 50 mM Tris-HCl, 2.5 mM EDTA, pH 7.0 with the same volume that was taken out after centrifugation. Resuspended cells were lysed by sonication using a microtip (6.4 mm) with 70% amplitude at 5 s On-time and 10 s Off-time for a total of 1 min On-time (4C15, BRANSON, Danbury, CT, USA). The lysed samples were centrifuged at 14,000 rpm for 10 min. The supernatant and inclusion bodies (IB) were separately collected and mixed with 4× Laemmli sample buffer with reducing agent 2-mercaptoethanol and boiled for 10 min. For SDS-PAGE and Western blot, 12% SDS-PAGE gels were used. After transfer onto a polyvinylidene difluoride membrane, the membrane was blocked overnight with 3% blotting-grade blocker in Tris-buffered saline with 1% Tween-20 (TBST) at 4 °C. After washing with TBST, the membrane was incubated for 1 h with anti-porcine circovirus antibody at 1:2,000,000, obtained from the Robinson lab at University of Kent, UK. The secondary antibody, Goat Anti-rabbit IgG H&L (Abcam, Waltham, MA, USA), was used at 1:10,000 dilution. The membrane was imaged on ChemiDoc™ (Bio-Rad, Hercules, CA, USA) with Western ECL Substrate. The Precision Plus protein dual color standard was used as a molecular weight marker. Protein concentrations were determined using Bradford assay. All reagents were sourced from Bio-Rad unless otherwise stated.

## 3. Results and Discussion

The Global Challenges Research Fund (GCRF) consortium has recently published a relatively inexpensive PCV2d vaccine candidate and a PCV2-PCV3 chimera vaccine candidate shown to effectively induce PCV2d- and PCV2-PCV3-neutralizing antibodies in immunized animals. In brief, their work showed the production of antisera to PCV2d and the neutralization effect of antisera with regard to PCV2d infection in cell-based assays. The virus-neutralizing (VN) titers were 28 ± 7 and 19 ± 2 for Rabbit 1 and Rabbit 2, respectively. The VN titers for pre-immunization sera were insignificant. This convinced them that the construct shows high potential for being further developed as a subunit vaccine against PCV2d, and hence that proceeding to large-scale process development should be carried-out [22].

Building on this work, the PCV2d subunit vaccine manufacturing process was further developed and verified at a pilot scale. Here, we present proof that the PCV2d vaccine candidate can be easily manufactured at a commercial scale. The process flow diagram is summarized in Figure 1. The workflow requires three steps: fermentation, cell disruption followed by filtration, and cation exchange chromatography. In-process controls, as indicated in the diagram, were included in each step, using analytical methods such as OD600, SDS-PAGE, Western blot, turbidity, and Bradford assay.

### 3.1. PCV2d Fermentation—From Bench Scale to Pilot Scale

At first, fermentation of PCV2d was attempted at a working volume of 1.5 L. Seed trains were prepared in 50 mL TB reaching an overnight OD600 of 10–12 and transferred into each fermenter where cells were cultured in chemically defined media (SM6Gc). Defined media provides benefits over other media, such as reducing batch-to-batch variability observed in complex media, increased process control for simple downstream processing, and lower vaccine production costs [23,24].

Batch fermentations were run for 24 h and yielded OD600 values of approximately 60 in batch 01 and 02, and 80 in batch 03 and 04. After this phase, the fed-batch was performed in constant mode at 0.15 mL/min. The calculated specific growth rates during the exponential phase of these four batches were in the range of 0.07–0.12 h^−1^. At the end of the fermentation, cell densities reached up to OD600 of 230, with the lowest cell density observed at OD600 of 170 (Figure 2). This variation between batches resulted in the calculated wet cell paste to culture weight ranging from 141 to 232 g/kg broth. This range was relatively wide, possibly due to the different liquid content in cell paste, although it is deemed acceptable for this application.

IPTG was added at sub-millimolar levels (0.039 g/L, equivalent to about 150 µM) to the fed-batch stage for induction of the *lac* promoter. The lower IPTG concentration was chosen to avoid toxicity and negative effect on cell growth [25,26]. PCV2d was well-expressed in both insoluble (inclusion body) and soluble fractions in all batches, as seen on the Coomassie gels with a band migrating at the expected molecular weight of 25 kDa for PCV2d (Figure 2B). These small-scale fermentations provided the operating conditions for the upstream process and demonstrated reproducibility in protein expression, despite variations in growth. The conditions were then applied for the development of a large-scale fermentation process.

To this end, and using the established bench scale parameters, the PCV2d vaccine was produced in large scale, demonstrating production capability in a 42 L stainless-steel fermenter with a 20 L initial working volume. The growth rate in large-scale fermentation was shown to be comparable to the 1.5 L fermentation runs shown in Figure 2A. During the exponential phase, the calculated µ was 0.15 h^−1^. Cells were grown to an OD600 of 220 corresponding to 6.59 kg wet cell paste with a total cultivation weight of 27.99 kg. This resulted in the calculation of 235 g wet cell paste per kg culture weight. Samples from various time points after IPTG induction were analyzed for PCV2d expression on SDS- PAGE and Western blot, as illustrated in Figure 2C. Soluble PCV2d was observed at 8 h post induction both in the insoluble and soluble fractions. The soluble fraction provided enough material for straightforward downstream processing and eliminated the need to isolate protein from inclusion bodies.

To illustrate the process transferability and reproducibility of this simple upstream process, large-scale production was carried out in parallel at the consortium partner site in the UK, as shown in Figure 2A (BL01 at KMUTT and BL02 at UCL). The cell cultures grew similarly with the calculated µ of 0.15 h^−1^.

### 3.2. Cell Disruption and Filtration

Next, PCV2d was purified from 60 g cell mass derived from large-scale fermentations. The cells were resuspended in 400 mL Tris-EDTA buffer and disrupted by homogenization in continuous mode. The homogenized pressures were varied between 700 and 2100 bar, and 700 bar was observed to be the optimal operating condition, as seen from the measurement of turbidity after centrifugation as well as the band intensity of protein release on the SDS-PAGE. The number of passages was set to 10, as the lowest viscosity of the mixture was observed with this number of passages. After centrifugation was carried out for liquid–solid separation, the supernatant fraction was then collected. A series of depth filtration and membrane filtration steps were then performed through 0.5–15 µm retention rating filters and 0.45 µm membrane filters. The turbidity of the filtered sample was measured at 215 NTU. Although turbidity seemed to be rather high, no untypical increase in column pressure was observed during sample application on the ion exchange column.

### 3.3. Cation Exchange Chromatography (CIEX)

The PCV2d protein has an isoelectric point (PI) of 9.4. Thus, a cation exchange column with bind/elute mode at pH 7.0 was selected for downstream purification. SP Sepharose Fast Flow has been widely used in preparative protein separations and is well known for high speed and low cost. Our preliminary studies (see Appendix A) with linear gradients from 25–90 mS/cm (200–700 mM NaCl) suggested that PCV2d started to wash out around 45 mS/cm (approximately 520 mM NaCl), as corresponding bands were observed on both SDS-PAGE and Western blot (see Appendix A). Thus, for our small-scale experiments with HiScreen columns, we optimized the binding conditions by increasing conductivity to improve the binding capacity of PCV2d while reducing impurities. We, therefore, increased the salt concentration to 25 mS/cm and 30 mS/cm, which is equivalent to 200 mM and 300 mM NaCl, respectively. The results shown in Figure 3 indicate that an experiment run without salt equilibration led to the early release of products in the flow through fractions (Figure 3A), whereas 200 mM or 300 mM NaCl concentrations in the equilibration buffer reduced impurities that were bound to the column, as we observed a smaller peak when using 200 mM NaCl (Figure 3B) and no peak when using 300 mM NaCl (Figure 3C) appearing at the wash step. As we observed impurities at 200 mM NaCl, we maintained the equilibration buffer at 300 mM NaCl. This optimization resulted in improved binding of PCV2d (by 17%) compared to the previously reported process [22]. Therefore, these conditions were applied to the HiScale 16/40 experiments, where the linear velocity was kept constant at 300 cm/h.

Here, 200 mL of clarified sample was loaded with a flow rate of 10 mL/min. The chromatogram shown in Figure 4A demonstrates only one peak at conductivity > 45 mS/cm, which was confirmed to be the product by SDS-PAGE (Figure 4B). In the first few fractions, PCV2d co-eluted with lower molecular weight proteins; later fractions contained the pure product. The elution fractions were pooled and the concentration was determined to be 0.94 mg/mL in 90 mL total volume (see Appendix A). This corresponds to a calculated yield of 18.6 g of pure PCV2d per one large-scale fermentation run or a yield of 0.93 g/L (18.6 g per 20 L). One full dose of the PCV2d vaccine requires 80 µg of PCV2d protein, as reported by Sno et al. [27]. The protein yield from our 20 L scale fermentation is therefore sufficient to vaccinate as many as 200,000 pigs with one dose.

## 4. Conclusions

We have presented a biomanufacturing process that is simple, transferable, and scalable to a large-scale protein subunit vaccine production for PCV2d. This process is based on a proof of concept of a novel PCV2d vaccine candidate that successfully induces PCV2d-neutralizing antibodies in immunized animals. All steps were carefully developed and evaluated to avoid excessive investment costs in low- and middle-income regions or high processing costs while maintaining high quantities for veterinary applications. We are optimistic that this developed process can benefit vaccine manufacturers, particularly in the region where the PCV2d vaccines are inaccessible. Additionally, we are convinced by our comparative studies between production sites that the presented PCV2d vaccine platform is highly likely to produce effective vaccines with high yields. However, other characterization parameters for veterinary vaccine products (i.e., endotoxin levels) may be required to meet the regulatory aspects. Further investigation in a pig’s field trial to assess the protection of our vaccine against a virulent PCV2d challenge may need to be considered.

## Figures and Tables

**Figure 1 vaccines-11-00169-f001:**
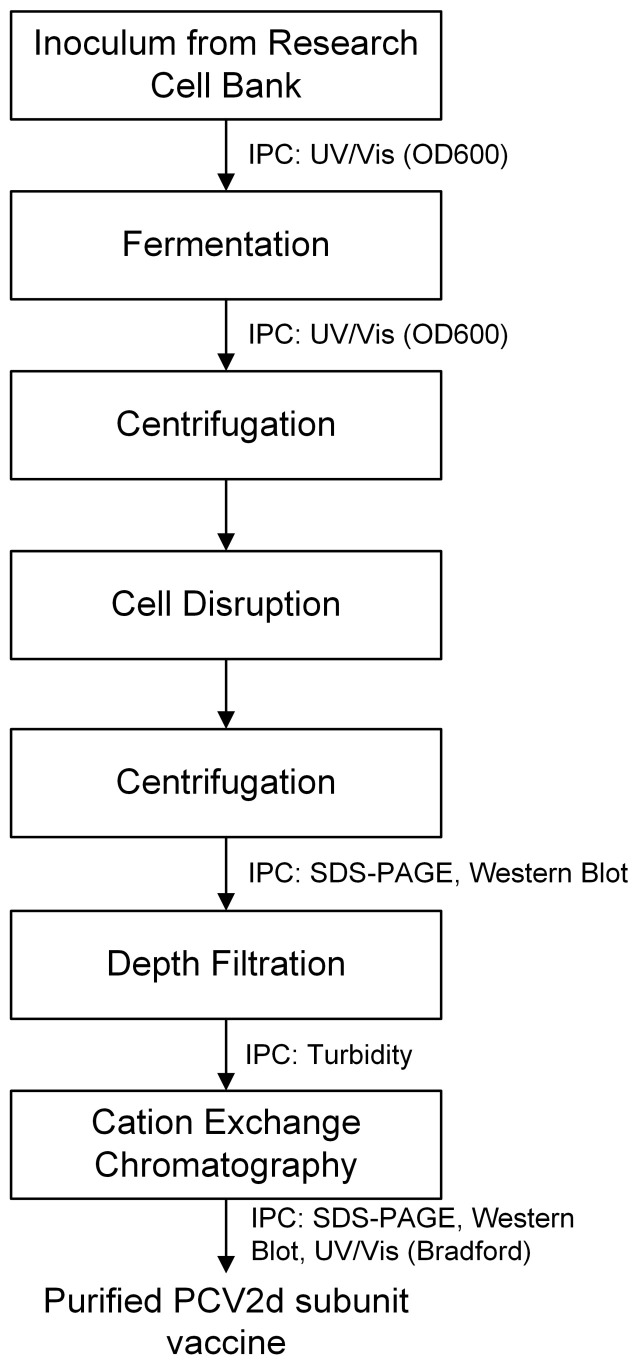
PCV2d process flow diagram. IPC = in-process control.

**Figure 2 vaccines-11-00169-f002:**
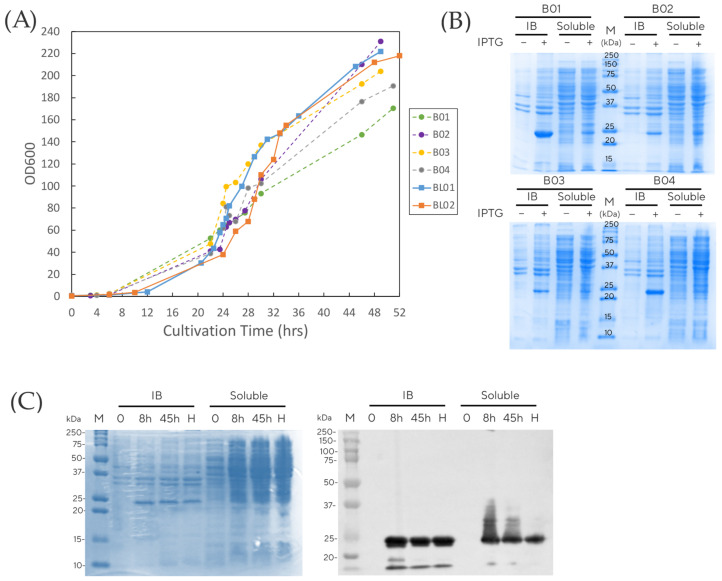
Fermentation results: (**A**) growth curve of *E. coli* producing PCV2d at bench scale (B01–B04) and pilot scale (BL01 and BL02), (**B**) SDS-PAGE analysis of PCV2d produced at bench scale, and (**C**) SDS-PAGE and Western blot analysis of PCV2d produced at pilot scale. IB = inclusion body, Soluble = supernatant, M = marker and H = harvest time.

**Figure 3 vaccines-11-00169-f003:**
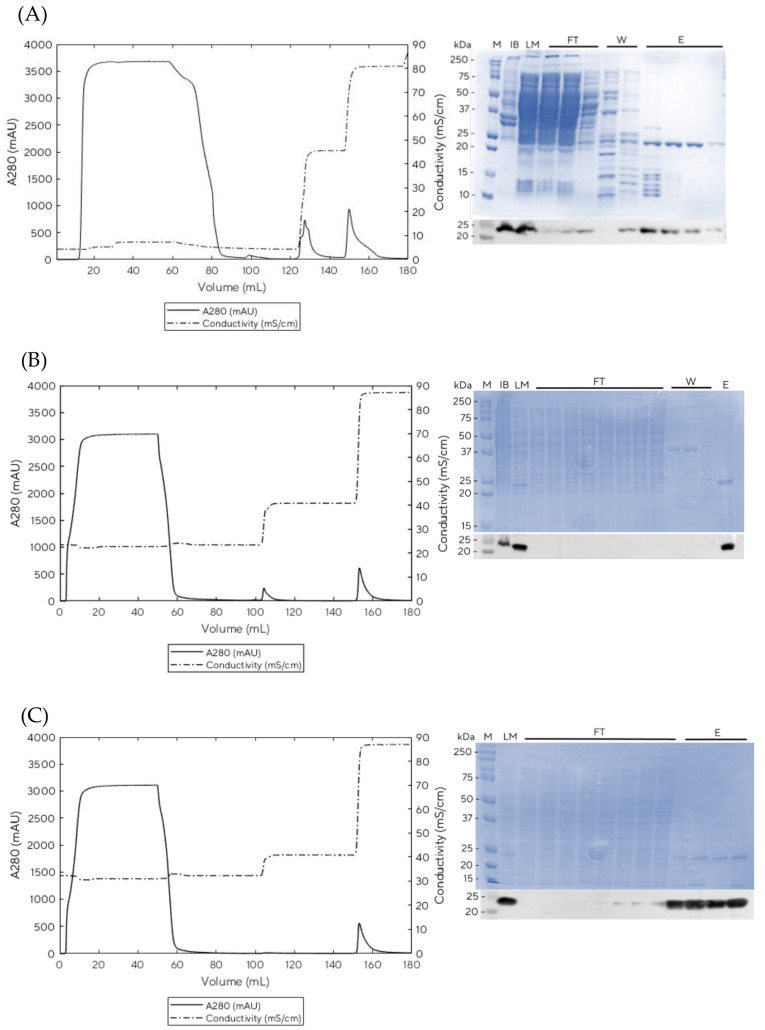
HiScreen SP Sepharose Fast Flow Chromatogram (left) and SDS-PAGE and Western blot analysis (right) of PCV2d purifications at various salt concentrations in equilibration buffer; (**A**) 0 mM NaCl, (**B**) 200 mM NaCl, and (**C**) 300 mM NaCl. M = marker, IB = inclusion body, LM = loading material, FT = flow through, W = wash and E = elution.

**Figure 4 vaccines-11-00169-f004:**
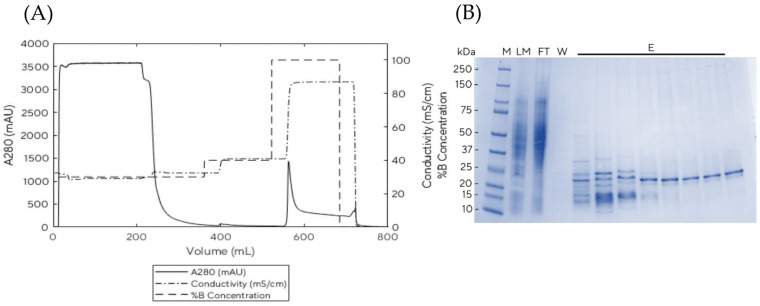
HiScale SP Sepharose Fast Flow Chromatogram (**A**) and SDS-PAGE analysis (**B**) of PCV2d purification. M = marker, IB = inclusion body, LM = loading material, FT = flow through, W= wash and E = elution.

## Data Availability

Not applicable.

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
