# Peer review of "A Simple and Cost-Efficient Platform for a Novel Porcine Circovirus Type 2d (PCV2d) Vaccine Manufacturing"

_vaccines, 2023, doi:10.3390/vaccines11010169_

Round 1

Reviewer 1 Report

Review of "A Simple and Cost-efficient Platform for a Novel Porcine Circovirus Type 2d (PCV2d) Vaccine Manufacturing"

This is a straight-forward manuscript that describes a relatively simple method for purifying a recombinant porcine circovirus protein (PCV2d-His6) from E. coli cells containing a pET23 vector in a stirred-tank pilot-scale fermenter.  The E. coli cells after induction with IPTG are collected by centrifugation, lysed using a high pressure homogenizer (10X) and removal of cell debris by high speed centrifugation.  The resulting supernatant fluid was filtered and subsequently subjected to cationic exchange column chromatography.  300 mM NaCl was determined to be optimal for column binding of the PCV-2d protein and elution from the column began at 520 mM NaCl.  SDS-page analysis (Fig. 4) indicated that a single band of the 25 kilodalton band (PCV-2d) was obtained in the eluate in the last 4 or 5 elution fractions.  This was also demonstrated in the Supplemental Figure 2. 

The investigators detected significant recombinant protein in the insoluble inclusion bodies removed in the high-speed centrifugation step and chose not to attempt to recover protein from this fraction. 

The investigators estimate that their yield of PCV2d protein from one large-scale fermentation batch of .93 g/L or 18.6 grams per the 20 L fermentation.  This does assume that 100% of the protein is in fact PCV2d and while the 25 kilodalton protein is the predominant band on a Coomassie stained gel, it does not mean that there are not a variety of minor contaminating proteins that contribute to this total protein figure.  It would be interesting to compare the Bradford value for total protein to an estimate of the protein amount on their 25 kilodalton protein band using known standard protein amounts by Coomassie staining.  My suspicion is that it is not 100% pure.

A couple of minor issues that the authors could address.

1.  A brief statement about why you chose not to use the His6 tag to purify the protein.  I suspect either cost of the nickel columns or lower yields (or both) are a likely answer.

2.  Endotoxin contamination.  For the PCV2d to be utilized for vaccination purposes, it will be necessary to control for endotoxin contamination in your preparations.  This may not be easily accomplished on the surface.  Most E. coli expressed recombinant proteins, even after purification, still have endotoxin residues.  I found a reference to the use of an E. coli LPS mutant strain in which the endotoxin is non-toxic (in humans) and not likely an issue (even though a likely contaminant).  The reference is "Detoxifying Escherichia coli for endotoxin-free production of recombinant proteins" Mamat et al., 2015 Microbial Cell Factories 14:57.  Using this E. coli strain could help to eliminate this issue for your PCV2d protein preparations for vaccination purposes.  A brief mention of the endotoxin issue in the Conclusions section could be important to avoid any issues with local laboratories producing their own vaccines.  

You are to be commended for offering this straight-forward protocol in such a detailed fashion that it could be reproduced in parts of the world that may not have ready access to commercial sources of PCV vaccines!

Author Response

Response to Reviewer 1 Comments
We are sincerely thankful for your time and yourinsightful comments. Please see below for our pointby-point response.
1. A brief statement about why you chose not to use the His6 tag to purify the protein. I suspect
either cost of the nickel columns or lower yields (or both) are a likely answer.
Response 1: Yes, the reason is the cost of the chromatography resin used in larger scale
manufacturing. We have preliminary results on various chromatography resins, and SP Sepharose
Fast Flow has been shown to yield good capacity binding to our PCV2d, better resolution in
separating impurities, and most importantly reasonable cost that we can easily do larger scale
production.
2. Endotoxin contamination. For the PCV2d to be utilized for vaccination purposes, it will be
necessary to control for endotoxin contamination in your preparations. This may not be easily
accomplished on the surface. Most E. coli expressed recombinant proteins, even after purification,
still have endotoxin residues. I found a reference to the use of an E. coli LPS mutant strain in which
the endotoxin is non-toxic (in humans) and not likely an issue (even though a likely
contaminant). The reference is "Detoxifying Escherichia coli for endotoxin-free production of
recombinant proteins" Mamat et al., 2015 Microbial Cell Factories 14:57. Using this E. coli strain
could help to eliminate this issue for your PCV2d protein preparations for vaccination purposes. A
brief mention of the endotoxin issue in the Conclusions section could be important to avoid any
issues with local laboratories producing their own vaccines.
Response 2: Thank you for your insightful comments. We agree that quality control for vaccination
is necessary, so we have added the following sentences in the conclusion section.
“We have presented a biomanufacturing process that is simple, easily transferrable, and expandable
to a large-scale E. coli platform producing protein subunit vaccines for PCV2d. All steps were
carefully developed and evaluated with the aim of avoiding excessive investment costs in low- and
middle-income regions or high processing costs while maintaining high quantities for veterinary
applications. Other characterization parameters for animal vaccine products (i.e. endotoxin levels)
may need to be considered in future to fulfill the regulatory requirements.”
You are to be commended for offering this straight-forward protocol in such a detailed fashion that
it could be reproduced in parts of the world that may not have ready access to commercial sources
of PCV vaccines!
Response 3: Thank you very much. We hope that this work will bring about a step-change in
countries like Thailand to be able to locally produce vaccines for the treatment of major human and
animal diseases in the future.

Reviewer 2 Report

The authors have developed a simple and cost-efficient platform for a novel PCV2d vaccine production, using fed-batch E. coli fermentation followed by cell disruption and filtration, and a single purification step via cation exchange chromatography. The conclusions and results were valuable for the vaccine of PCV2d. However, there were some issues in the manuscript. 

Line 18 E.coli should be italic.

Line 41: The “,” should be “.”

Line 44: The first abbreviation “E.coli” should be given the complete name.

Line 65: The “OD600” should be revised as “OD600nm”. The throughout paper should be changed.

Line 89: The numbers of “H3PO4, NH4OH” should be subscript.

Line 90 to 92: The sentence should be changed and it is too long.

Line 190-191: The error should be changed. The throughout paper should be changed for the reference errors.

The TEM should be performed and the picture of VLP should be shown.

In Figure 2, the (B) and (C): The backgrounds of SDS-PAGE should be same.

The figure 1 should be as supplementary file.

Author Response

Response to Reviewer 2 Comments
Thank you for your comments. We greatly appreciate the reviewer’s insightful suggestion and we
have revised the manuscript to address your concerns. Please see our response point by point.
Extensive editing of English language and style required
Response: the English language and style have been edited by using a writing assistant tool,
Grammarly, as well as having co-author Dr Stefanie Frank proofreading.
Are all the cited references relevant to the research? – Must be improved
Response: we have thoroughly reviewed the cited references and they are relevant to the research.
Are the methods adequately described? – Must be improved
Response: we have improved the description of the methods for better clarification by changing
several long sentences to shorter ones. We hope it is now clearer and more adequate.
Line 18: E.coli should be italic.
Response: it has been revised to italic.
Line 41: The “,” should be “.”
Response: we have corrected it.
Line 44: The first abbreviation “E.coli” should be given the complete name.
Response: we have included the complete name of E. coli as suggested.
Line 65: The “OD600” should be revised as “OD600nm”. The throughout paper should be changed.
Response: we use OD600 according to our Eppendorf device’s white paper (the references are
shown below). OD600 is also widely used in publications and here are some examples;
Janke S A, Fortnagel P, Bergmann R. Microbiological turbidimetry using standard photometers.
Biospektrum, 1999, Vol. 6, 501-502.
White Paper; “Factors Influencing OD600 Measurements – Which factors influence microbial
growth and with this varying absorbance values of turbidity measurements using the same
photometer?”, Eppendorf AG, 2015
Beal, J., Farny, N.G., Haddock-Angelli, T. et al. Robust estimation of bacterial cell count from optical
density. Commun Biol 3, 512 (2020). https://doi.org/10.1038/s42003-020-01127-5
Munakata, Y.; Heuson, E.; Daboudet, T.; Deracinois, B.; Duban, M.; Hehn, A.; Coutte, F.; SlezackDeschaumes, S. Screening of Antimicrobial Activities and Lipopeptide Production of Endophytic
Bacteria Isolated from Vetiver Roots. Microorganisms 2022, 10, 209.
https://doi.org/10.3390/microorganisms10020209
Line 89: The numbers of “H3PO4, NH4OH” should be subscript.
Response: we have revised the numbers to be subscript.
Line 90 to 92: The sentence should be changed and it is too long.
Response: instead of having long sentences, we have revised them to the following sentences.
“Dissolved oxygen (DO) concentration (%DO) was calibrated at 0 by unplugging the probe’s cable.
Calibration of 100%DO was achieved by keeping the mixing speed at 1050 rpm and the airflow rate
at 60 Nl/h. The %DO setpoint was 30 and was controlled by a cascade of agitation speeds (450 -1050
rpm), airflow, and O2 flow. A total gas flow of 1 vvm (volume of gas sparged per fermenter
working volume per minute) was used.”
Line 190-191: The error should be changed. The throughout paper should be changed for the
reference errors.
Response: we apologise for our mistakes in referencing Figures. We have now revised the errors.
The TEM should be performed and the picture of VLP should be shown.
Response: The PCV construct was from Robinson group where they do not expect the construct to
form a VLP as the first 40 amino acids, which are shown to promote the VLP formation, have been
deleted. Their PCV2d study on size-exclusion chromatography shows no evidence of VLP
formation. Here is the reference.
Peswani, A.R.; Narkpuk, J.; Krueger, A.; Bracewell, D.G.; Lekcharoensuk, P.; Haslam, S.M.; Dell, A.;
Jaru-Ampornpan, P.; Robinson, C. Novel constructs and 1-step chromatography protocols for the
production of Porcine Circovirus 2d (PCV2d) and Circovirus 3 (PCV3) subunit vaccine candidates.
Food and Bioproducts Processing 2022, 131, 125-135, doi:10.1016/j.fbp.2021.10.001.
In Figure 2, the (B) and (C): The backgrounds of SDS-PAGE should be same.
Response: we have revised the figure as suggested.
The figure 1 should be as supplementary file.
Response: as the main message of our manuscript is the platform manufacturing that is simple,
cost-effective and transferable so we think the overview of the process as shown in Figure 1 is
needed in the main text.

Round 2

Reviewer 2 Report

The authors have bee revised the manuscript based on my suggestions.